# Thin Films of Tolane Aggregates for Faraday Rotation: Materials and Measurement

**Maarten Eerdekens** [1] **, Ismael López-Duarte** [2] **, Gunther Hennrich** [2,*] **and Thierry Verbiest** [1,*]

[1]   Department of Chemistry, University of Leuven, Celestijnenlaan 200D, 3001 Leuven, Belgium;
      maarten.eerdekens@kuleuven.be
[2]   Department of Organic Chemistry, Universidad Autónoma de Madrid, Cantoblanco, 28049 Madrid, Spain;
      ismael.lopez@uam.es
*    Correspondence: gunther.hennrich@uam.es (G.H.); thierry.verbiest@kuleuven.be (T.V.)

**Abstract:** We present organic, diamagnetic materials based on structurally simple (hetero-)tolane derivatives. They form crystalline thin-film aggregates that are suitable for Faraday rotation (FR) spectroscopy. The resulting new materials are characterized appropriately by common spectroscopic (NMR, UV-Vis), microscopy (POM), and XRD techniques. The spectroscopic studies give extremely high FR activities, thus making these materials promising candidates for future practical applications. Other than a proper explanation, we insist on the complexity of designing efficient FR materials starting from single molecules.

**Keywords:** faraday rotation; thin films; magneto-optics; organic material; tolane derivatives

## 1. Introduction

Faraday rotation (FR) is a magneto-optic (MO) effect that was discovered more than a century ago [1]. It is the rotation of the plane of polarization in the presence of a longitudinal magnetic field, and the rotation angle θ can be described by θ = *VBL* with the angle of polarization rotation, V the Verdet constant, *B* the magnetic field parallel to the propagation of light, and *L* the path length. Applications of FR are of practical relevance for magnetic field sensors, wave guiding, fiber-optics, etc. [2–4]. Traditionally the field of magneto-optics has been dominated by inorganic materials or radical species [5–8]. Only recently have diamagnetic organic materials emerged as novel FR supplies [9–14].

Although the exact origin of Faraday rotation in organic molecules is currently unknown, different research groups have dedicated their efforts to designing new organic materials for FR applications. Current experiments reported in the literature clearly suggest that molecular conjugation and π-stacking are crucial factors to obtain very strong FR. Furthermore, for organic diamagnetic materials, it became evident that the macroscopic order of the bulk material is crucial for its optical and MO activity [9–14]. It is this duality of molecular vs. macroscopic material, i.e., intra- vs. intermolecular processes, that complicates a rational correlation of the observed magnetic effects with the nature of molecular units and supramolecular aggregates. We have recently shown how the structural simplification of the molecular units (from trigonal to linear) has led to an increase in the FR activity of the respective thin-film materials [15]. Nonetheless, one has to keep in mind the macroscopic structure of the respective aggregates. A decisive requirement is the capacity to form quality thin films from molecular units or aggregates, either crystalline [16] or liquid crystalline [17]. It was shown that long-range electron movement along columnar supramolecular aggregates leads to a dramatically increased Faraday response [18].

## 2. Materials and Methods

In the present work, we present tolane structures that form crystalline thin films and assess their MO activity by FR spectroscopy (Figure 1).

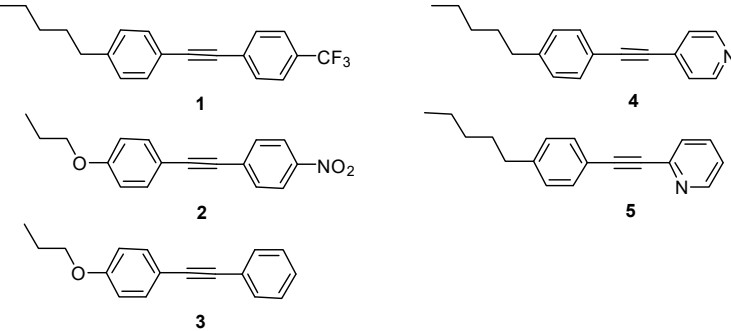

**Figure 1.** Tolanes (**1**–**3**) and *N*-Hetero-tolane derivatives **4** and **5**.

The diphenyltolanes **1**–**3** present a conventional donor-π-acceptor system. Compounds **2** and **3** have been studied previously for their second-order NLO properties [19].

### 2.1. Bulk Properties

Tolane 1 was synthesized following the literature procedure and was obtained in a 72% yield [20]. Characterization: $^1$H NMR (400 MHz, CDCl$_3$) $\delta_H$ 7.56 (m, 4H), 7.47 (d, *J* = 8.1 Hz, 2H), 7.17 (d, *J* = 8.1 Hz, 2H), 2.60 (t, *J* = 7.5 Hz, 2H), 1.61 (q, *J* = 7.5 Hz, 2H), 1.31 (m, 4H), 0.91 (t, *J* = 6.8 Hz, 3H); $^{13}$C NMR (100 MHz, CDCl$_3$) $\delta_C$ 144.5, 131.9, 131.9, 130.5, 129.6, 128.7, 127.6, 125.4, 119.9, 92.3, 87.6, 36.1, 31.6, 31.0, 22.7, 14.1. EI$^+$-MS m/z 316 (M+); Anal. calcd. for: C$_{20}$H$_{19}$F$_3$: %C, 75.93; %H, 6.05; found: %C, 75.88; %H, 5.97.

Differential scanning calorimetry (DSC) analysis revealed the existence of two main crystalline polymorphs melting at 69.7 and 71.3 °C, respectively. These transitions stabilized after two heating–cooling cycles (Figure 2).

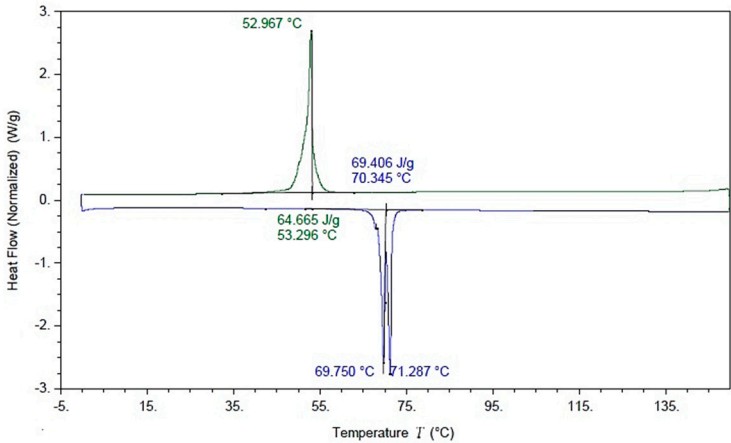

**Figure 2.** Differential scanning calorimetry (DSC) of **1**, second heating (green) and cooling (blue) cycle.

The *N*-pyridyl-tolanes **4** and **5** were solid (**4**) or liquid (**5**) at room temperature, respectively. However, upon protonation (with hydrochloric or terephthalic acid), both formed crystalline solids. In addition, solid halogen bond complexes were obtained from **4** and **5** with suitable halogen bond donors. The terephthalate complex of protonated **4** was liquid crystalline over a wide temperature range. Hence, the material can be processed and measured conveniently in a conventional liquid crystal (LC) cell.



## 2.2. Spectroscopy and Thin Film Preparation

UV-Vis Spectra were measured in chloroform at a concentration of $1.5 \times 10^{-4}$ mol/L using a Perkin-Elmer Lambda 900 spectrophotometer (Norwalk, CT, USA). To measure the Faraday rotation as well as polarized optical microscopy (POM) [21], the materials were placed in LC cells with a 3 μm gap. To fill the cells with the organic materials, a heating plate heated the cells to a temperature of 5 to 10 °C higher than the melting temperature of the desired molecules. A small amount of material was deposited next to the gap. It subsequently melted, and entered the cell through capillary action. After the LC cell had been filled and cooled, a homemade heating and cooling device reheated the filled LC cells to 5 °C above the melting temperature of the organic material, and cooled the samples to r.t. by 0.1 °C/min. An Olympus microscope was used for obtaining POM images. Faraday rotation spectra were collected using a photoelastic modulation magneto-optical setup described by Vandendriessche et al. from 350 to 700 nm, every 2 nm [4]. The optical rotation was measured at varying magnetic field from 0 to 0.5 T. A blank was also measured to nullify the effects of the glass. Using linear regression, the magnetic rotation was calculated from the slope. The Verdet constant (°/Tm) was then calculated by dividing the magnetic rotation by the thickness of the sample inside the LC cell, i.e., 3 μm. Smoothing of the curves was done using Savitsky–Golay method in Origin. We confirmed that the sample was in the plane was isotropic by measuring at different azimuthal angles of the samples at 400 nm (Figure S1).

## 3. Results and Discussion

Compounds **1–4** were crystalline solids at room temperature. The powder X-ray diffractograms were measured on a Malvern PANalytical Empyrean system, with a Cu K-$\alpha$ source with a wavelength of 1.5406 Å, measuring with a PIXcel3D detector (Malvern analytical, Eindhoven, Noord-Brabant, The Netherlands). The measurements were done at room temperature. The XRD (and POM) measurements confirmed the crystal nature of the samples (Figure S2).

The absorbance spectra of **1** and **2** showed a maximum absorbance of around 350 nm with a corresponding high FR of several hundred thousand °/Tm. This is not surprising since the FR response is enhanced near resonances. However, what is surprising is that even far away from resonance strong Faraday rotation was observed. For example, for compound 1, the Verdet constant in the wavelength region 525 to 700 nm nanometers was still on the order of 50,000 to 70,000°/Tm, while compound 2 exhibits Verdet constants over 150,000°/Tm around 500 nm. Similar behavior has been observed for other crystalline acetylenes [5]. Both molecules had an electron acceptor group ($-NO_2$ and $-CF_3$) within a conjugated π-system. They showed very similar FR spectra with several peaks and valleys in the visible part of the spectrum (Figure 3A). Tolane 2 exhibited a higher Faraday response over most part of the spectrum (Figure 3B).

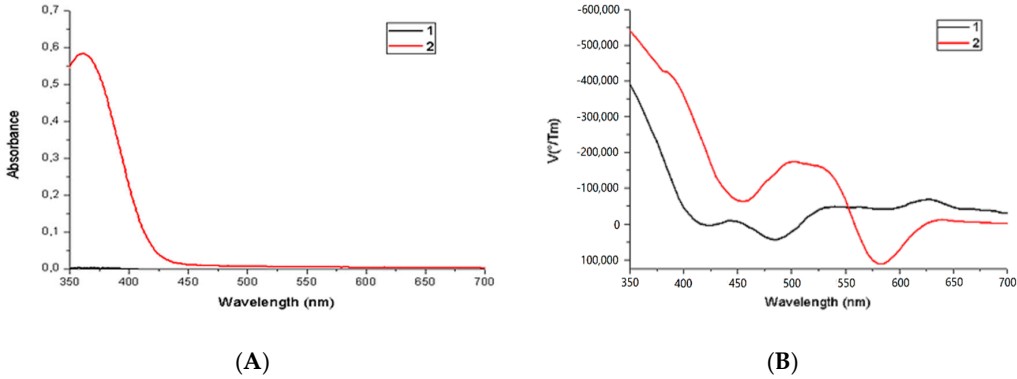

**Figure 3.** (**A**) and Faraday rotation spectrum (**B**) of **1** and **2**.

Our results for compounds **4** and **5**, nicely illustrate the importance of their macroscopic structure. Compound **5** was an isotropic liquid at r.t. temperature, while compound 4 was solid at room temperature. Therefore and in agreement with earlier findings, **4** exhibited a Faraday response that was orders of magnitude higher over the entire °/Tm, even outside regions of absorption. For 5, we measured a featureless and low-intensity Faraday spectrum that gradually decreased towards the IR region (Figure 4B). Note also that the Faraday spectrum of 4 resembles that of **1** and **2** with a maximum Verdet constant near resonance and a very feature-rich shape in the visible part of the spectrum. This seems to indicate that molecular structure does have an impact on the shape of the Faraday spectrum. Moreover, the supramolecular organization (either in a crystalline or liquid form) is a necessary requirement to observe strong FR activity.

Of all the samples, the strong increase in FR towards the UV part of the spectrum was due to the presence of the absorption band as well as the wavelength dependency of the Verdet constant ($V \sim \lambda^{-2}$). We do not know the origin of the peaks and valleys, but recent work on Faraday rotation in other organic molecules suggests crediting them to the presence of spin-forbidden or hidden singlet and/or triplet states [4]. The non-substituted diphenylacetylene 3 exhibited too much birefringence and scattering to perform reliable measurements. Its UV-Vis absorption spectrum can be found in the supporting information (Figure S3). POM images can be found in the supporting information (Figure S4).

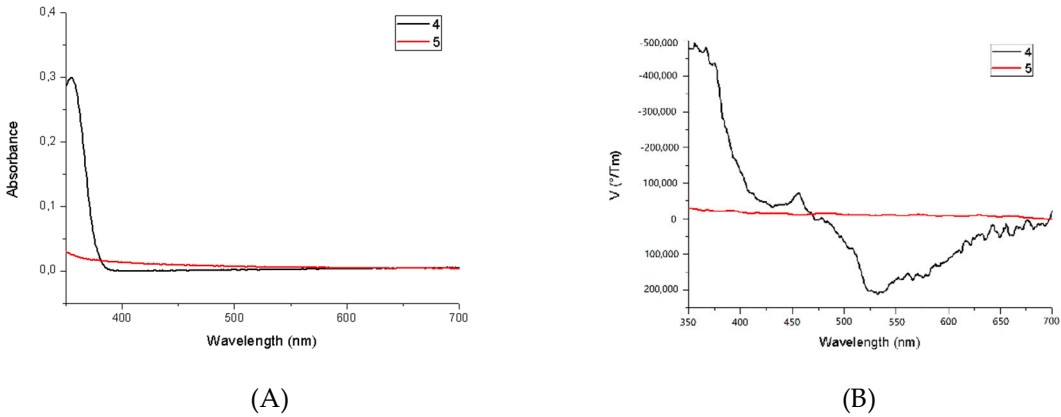

(A)　　　　　　　　　　　　　　　　　　　　　　　　　　　　(B)

**Figure 4.** UV-Vis (**A**) and Faraday rotation spectrum (**B**) of 4 and 5.

## 4. Conclusions

We have investigated the FR response of different phenylacethylene derivatives. It is clear that macroscopic organization within the bulk material is a key factor in obtaining a high Faraday response. The molecular structure dictates the macroscopic organization of the building units, which, in consequence, determines the FR response of the resulting bulk material. The detailed shape of the Faraday spectrum is a result of this (i.e., the molecular structure), but is in itself not sufficient to create a high Faraday response. Once again, we have to emphasize the aforementioned duality (single molecule vs. macroscopic bulk) that interferes with the design of efficient FR materials.

The tolanes that were crystalline at room temperature showed very high Verdet constants—much higher than typically observed for diamagnetic materials—in regions of the spectrum where there is no optical absorption, making them potentially useful for applications.

**Supplementary Materials:** The following are available online at http://www.mdpi.com/2079-6412/9/10/669/s1, Figure S1: Verdet constant measurements at 400 nm: Verdet constant of samples 1, 2, 4 and 5 turned azimuthal 0°, 30°, 60° and 90°. The Verdet constant was measured at 400 nm. No dependence of Verdet constant on the rotation of azimuthal angle was observed; Figure S2: X-ray diffractograms: tolanes (1–3) and n-hetero-tolane derivatives 4; Figure S3: UV-Vis absorbance spectrum of the unsubstituted diphenylacetylene (3); Figure S4: Polarized optical microscopy: polarized optical microscopy images of the materials in the LC cells.

**Author Contributions:** Conceptualization, T.V. and G.H.; methodology, M.E.; validation, M.E.; formal analysis, M.E.; investigation, M.E.; Resources: LC: T.V. and M.E., Synthesis, I.L.-D. and G.H.; data curation, M.E.; writing—original draft preparation, M.E. and T.V.; writing—review and editing, M.E., T.V. and G.H.; visualization, M.E. and G.H.; supervision, M.E. and G.H. and T.V., project administration, M.E. and G.H.; funding acquisition, G.H and T.V. All authors have given approval to the final version of the manuscript.

**Funding:** This research was funded by the Spanish government (MINECO, project CTQQ2016-7557-R) and the KULeuven (C1 project).

**Acknowledgments:** The authors would like to thank B. Goderis for using his POM equipment and M. Rouffaers and L. Clinckemalie for the X-ray data analysis.

**Conflicts of Interest:** The authors declare no conflict of interest.

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
