# Peer review of "Thin Films of Tolane Aggregates for Faraday Rotation: Materials and Measurement"

_coatings, doi:10.3390/coatings9100669_

Round 1

Reviewer 1 Report

Generally, the report on the Faraday rotation of diamagnetic materials is important. However, the measurement of FR spectra of solid materials require some careful consideration. For example, the solid material may have intrinsic heterogeneity inside. Therefore, the measurement has to be done by rotating the sample cell. It is not clear that such measurement was done in this study.

The reported FR spectra has to be discussed by assigning the maximum bands.  

Author Response

Reply to Reviewer 1:

We confirmed the isotropy of the sample by turning the sample asmuthal. The measurements need to be put in the Supporting information. Extra line was added with materials and methods (line 73-75):

“We confirmed that the sample is in plane isotropic by measuring at different amutal angles of the samples (SI)”

- The Following graph with caption should be added to the supporting information.

Figure xx: Verdet constant of samples 1,2,4 and 5 turned asmuthal 0°, 30°, 60° and 90°. The verdet constant was measured at 400 nm. No dependence of Verdet constant on rotation of asmuthal angle was observed.

“The reported FR spectra have to be discussed by assigning the maximum bands.”

We added following explanation (line 115 – 120):

“The strong increase in FR towards the UV part of the spectrum is due to the presence of the absorbtion band as well the wavelenght dependecy of the verdet constant (). We do not know the origin of  te peaks and the valleys, but recent work on Faraday rotation in other organic molecules suggests crediting them to the presence of spin-forbidden or hiddensinglet and/or triplet states[4]. “

Reviewer 2 Report

The authors report on a series of diphenyltolanes with different electronic structure which present very high Verdet constants, even far away from resonance. The results of  this study do not drive to clear structure-activity requirements at a molecular level, but reinforce the necessity that organic materials have a  long range order in order to exhibit high Faraday rotation  activity. Considering the great  interest that materials with high magneto optical  activity arouse in applications such as sensing or communication technologies and the scarcity of organic materials that present this property, I consider that this manuscript is of interest for  Coatings.

The following minor revisions should be addressed. 

-The manuscript  should be thoroughly revised. Many typos can be found throughout the text.

-Some details on the synthesis of the new compounds should be included in the main  body of the manuscript.

-X-ray diffractograms should be moved to the supporting information, as their different patterns are not a fundamental part in the discussion.

Author Response

Reply to Reviewer 2:

- Typos are corrected.

- The synthesis of compound 1 is included in the text body:

2, l. 46: ¨Tolane1is synthesized following a standard Sonogashira protocol. Commercial4-ethyny-1- pentyl benzene and trifluoromethyl benzene in equimolar amounts under Pd(Ph3)2Cl2/CuIcatalysisare reacted accordingly to give 1 in 72% yield.Characterization: 1H NMR (400 MHz, CDCl3) δH7.56 (m, 4H), 7.47 (d, J = 8.1 Hz, 2H), 7.17 (d, J = 8.1 Hz, 2H), 172 2.60 (t, J = 7.5 Hz, 2H), 1.61 (q, J = 7.5 Hz, 2H), 1.31 (m, 4H), 0.91 (t, J = 6.8 Hz, 3H); 13C NMR (100 MHz, 173 CDCl3) δC144.5, 131.9, 131.9, 130.5, 129.6, 128.7, 127.6, 125.4, 119.9, 92.3, 87.6, 36.1, 31.6, 31.0, 22.7, 14.1. EI+-174 MS m/z 316 (M+); Anal. calcd. for: C20H19F3: % C, 75.93; % H, 6.05; found: % C, 75.88; %H, 5.97.

- Change reference on p. 7, 172-175 for: K. Sonogashira. Development of Pd–Cu catalyzed cross-coupling of terminal acetylenes with sp2-carbon halides. J. Organomet. Chem. 2002, 653, 46 - 49.

- Move XRDs from the text body to the supporting information.

Reviewer 3 Report

The article refers to one of the hot issues of modern nanoscale physics – creation of new effective magneto-optical materials and is a continuation of previous studies of the authors on the Faraday effect in thin-film structures based on organic molecules. Here, they have suggested several new structures with high Faraday effectiveness. I would like to recommend the article for publication, but before authors should edit the manuscript carefully.

The statement of the aim of this work is desirable. At least, text from lines 39, 40 in page 1 should be rearranged directly in the paragraph 1 immediately after line 37. Subtitle 2.1 should be deleted since the text following it describes the synthesis method, not massive properties. The next subtitle 2.2 also seems to be redundant. Figures showing the XRD patterns have no numbers and figure captions and are not described in the text. This must be corrected and, accordingly, the numbering of subsequent figures. Maximum absorbance around 350 nm is seen only for sample 2. The citation order is not convenient for readers. Each Reference should have its own number and this number should be indicated in the text. Several unclear places in the manuscript is colored in the pdf file attached.

Author Response

Reply to Reviewer 3:

- p1. Rephrased: ¨It is the rotation of the plane of polarization in the presence of a longitudinal magnetic field. - - The rotation angle q can be described by q =VBL with q the angle of polarization rotation,…¨

- l. 30: ¨magneto-optic (MO)¨…

- p. 2, line 44: instead of ¨NLO¨: second order nonlinear optical¨…

- subtitle 2.1. is deleted.

- Subtitle 2.2. is deleted

-XRD see reviewer 2 corrections.

- Regarding the attatched pdf document (very helpful!), changes have been made on P.1, l19; 30; 78, 100.

- The citation has been revised and changed as stated earlier (ref. [10]).

- Rev.3 also addresses the XRD images which have been moved.

-On p.6, l34 the funding information needs to be changed to CTQ2016-7557-R

-  l 135 Communidad de….Should be deleted.

- The manuscript has been revised for typos.

Round 2

Reviewer 1 Report

Some corrections about the words are required before acceptance for publication.

Line 76, simple, iso tropic, amutal